# The STAT3/Slug Axis Enhances Radiation-Induced Tumor Invasion and Cancer Stem-like Properties in Radioresistant Glioblastoma

**DOI:** 10.3390/cancers10120512

**Published:** 2018-12-13

**Authors:** Jang-Chun Lin, Jo-Ting Tsai, Tsu-Yi Chao, Hsin-I Ma, Wei-Hsiu Liu

**Affiliations:** 1Graduate Institute of Clinical Medicine, College of Medicine, Taipei Medical University, Taipei 11042, Taiwan; 13451@s.tmu.edu.tw (J.-C.L.); oncojerry@tmu.edu.tw (T.-Y.C.); 2Department of Radiation Oncology, Shuang Ho Hospital, Taipei Medical University, Taipei 23561, Taiwan; kitty4024@gmail.com; 3Department of Radiology, School of Medicine, College of Medicine, Taipei Medical University, Taipei 11042, Taiwan; 4Division of Hematology/Oncology, Shuang-Ho Hospital, Taipei Medical University, Taipei 23561, Taiwan; 5Department of Neurological Surgery, Tri-Service General Hospital and National Defense Medical Center, No. 325, Sec. 2, Cheng-Kung Road, Taipei 11490, Taiwan; uf004693@mail2000.com.tw; 6Department of Surgery, School of Medicine, National Defense Medical Center, Taipei 11490, Taiwan

**Keywords:** glioblastoma, STAT3, slug, radioresistance, cancer stem cell (CSC)

## Abstract

Glioblastoma multiforme (GBM) requires radiotherapy (RT) as a part of definitive management strategy. RT is highly effective, destroying cancer cells that may exist around the surgical tumor bed. However, GBM still has a poor prognosis and a high local recurrence rate after RT. Accumulating research indicates that GBM contains cancer stem-like cells (CSCs), which are radioresistant and result in therapeutic failure. Additionally, GBM cells can aggressively invade normal brain tissue, inducing therapeutic failure. Using clinical observations, we evaluated the effect of radiation on tumor control. We also explored the biomolecular pathways that connect radioresistance and CSC- and epithelial-mesenchymal transition (EMT)-associated phenotypes in patient-derived GBM cells. Transwell and microarray assay demonstrated that radioresistant GBM cells (GBM-R2I2) exhibit increased invasion and self-renewal abilities compared with parental GBM cells. Finally, to identify potential mechanisms underlying these observations, we used a PCR array to search for molecular markers of cell motility. Signal transducer and activator of transcription 3 (STAT3) directly bound to the Slug promoter in a chromatin immunoprecipitation assay. Reduced STAT3 decreased Slug expression and suppressed cell invasion in GBM-R2I2 cells while increasing Slug reversed these effects. In addition, STAT3 knockdown significantly inhibited CSC properties, synergistically increased the radiotherapeutic effect, and effectively increased the survival rate in vivo. We deciphered a new pathway of GBM radioresistance, invasion, and recurrence via the STAT3/Slug axis that could be a new target of GBM therapy.

## 1. Introduction

Glioblastoma multiforme (GBM), malignant glioma of World Health Organisation (WHO) Grade IV [1], is the most devastating brain malignancy because it is resistant to all modern treatments, such as surgery radiotherapy (RT), chemotherapy, and immunotherapy [2]. GBM recurs frequently after transiently exhibiting complete remission by conventional imaging, indicating that resistant glioma cells may be undetectable by such imaging. These cells have the ability to regrow the primary tumor and thus promote recurrent disease [3].

Irradiated GBM cells have been thought to act as cancer stem-like cells (CSCs) with a high self-renewal capacity, relative quiescence, and protection by the niche, thus underlying tumor recurrence and radioresistance [4,5]. Human irradiated GBM specimens have been found to be enriched in CSCs [6]. Infiltration of malignant glioma cells into the normal brain parenchyma is a prominent feature of GBM and makes complete surgical resection almost impossible. Accumulating studies have indicated a critical role of epithelial-mesenchymal transition (EMT) during GBM progression and its association with increased tumor invasion. Tumor progression is accompanied by increasing invasiveness in GBM [7]. Sublethal irradiation promotes cell migration and invasiveness of glioma cells [8]. Moreover, fractionated ionizing radiation (IR), which is similar in use to clinical RT, enhances the portion of the CSC population and migration of GBM cells in vivo [9,10]. However, it is still not understood whether the enhancement of such mechanisms is an inherent component of CSC adaptation and tumor motility in response to repeated radiation.

Signal transducer and activator of transcription 3 (STAT3) regulates cell invasion, cell cycle progression, and apoptosis [11]. STAT3 activation in glioblastoma is a poor prognostic factor for tumor growth and survival rate [12]. Suppression of STAT3 expression effectively reduces GBM tumor growth [13]. Several studies have shown that STAT3 regulates the EMT process in many types of cancers [14,15]. EMT promotes a triad of tumor features: Stem cell properties, tumor invasion, and radioresistance [16]. The constitutive activation of STAT3 pathways leads to the regulation of glioma CSCs [17]. The importance of STAT3 in modulating radioresistance has recently been emphasized in GBM [18]. It is unclear whether STAT3 activation is involved in the acquisition of cancer stem-like cells and EMT-like properties, such as tumor invasion, to modulate the radiosensitivity of GBM.

Accumulating evidence has highlighted the molecular mechanisms underlying GBM cell migration and tumor invasion. Some EMT transcription factors, including Slug and Snail, have been shown to enhance radioresistance by antagonizing p53-mediated apoptosis [19]. CSCs derived from GBM, demonstrate greater tumor invasive potential in vitro and in vivo than matched non-stem glioma cells [20]. To date, the correlations of EMT with CSCs and its associated processes in GBM have received little attention, but it is clear that important CSC-like properties/invasion pathways overlap between the central nervous system and cancers [16,21]. In our study, we established GBM-derived irradiated/invasive cells (R2I2) as radioresistant GBM cell lines and hypothesized a novel relationship between STAT3 and Slug expression in R2I2 cells to regulate CSC properties, tumor invasion, and radioresistance. Our confirmation of this hypothesis suggests a crucial process contributing to disease recurrence in the clinic.

## 2. Methods

### 2.1. Patient Characteristics

Patients with malignant glioma were treated for primary brain tumors and perifocal edema using methods approved by the multidisciplinary central nervous system (CNS) tumor board at Shuang Ho Hospital and Tri-Service General Hospital, National Defense Medical Center. The present study was approved by the Institutional Review Board Committee of Tri-Service General Hospital, National Defense Medical Center (TSGHIRB No. 1-107-05-107, 28 June 2018).

### 2.2. Cell Culture

We collected patient-derived glioblastoma cell lines in accordance with the tenets of the Declaration of Helsinki, and all protocols were approved by Institutional Review Committee. Cell culture procedures were performed according to previous methods [22].

### 2.3. RT^2^ Profiler PCR Array

The RT2 Profiler PCR Array of Human Epithelial to Mesenchymal Transition, which detects the expression of 84 key genes, was purchased from SABiosciences (Frederick, MD, USA). Fold changes in GBM relative to control samples were calculated by the manufacturer for the expression of genes related to cell motility and EMT (RT2 Profiler PCR Array Data Analysis Template v3.3, Qiagen, Valencia, CA, USA).

### 2.4. Quantification of the Speed of Motile Cells

Cell motility was observed by time-lapse microscopy as previously described [23]. We quantified the speed of the motile cells according to previously described methods [24].

### 2.5. Plasmids and Gene Transduction

STAT3- and Slug-overexpression plasmid construction and DNA preparation followed standard procedures for the Trans-Messenger Reagent Kit (Qiagen, Valencia, CA, USA). The genes encoding human STAT3 (NM_003150) and human Slug (NM_003068) were cloned by previously described methods [24].

The shRNA targeting human STAT3 (NM_003150, 5′-GCACAATCTACGAAGAATCAA-3′) and human Slug (NM_003068, 5′-ATGCTCATCTGGGACTCTGTC-3′) were cloned by previously described methods [24].

### 2.6. Chromatin Immunoprecipitation (ChIP) and Q-ChIP Assay

For ChIP analysis, cells were cotransfected with empty vectors or STAT3-expressing vectors combined with various constructs of Slug promoter regions, as indicated in the Results section. Immunoprecipitation reactions were performed by previously described methods [24]. Quantitative Real-Time Reverse Transcription results were normalized to Immunoglobulin G (IgG) internal controls. Appendix A presents the sequences of the primers used for the Quantitative Chromatin Immunoprecipitation (ChIP) experiments.

### 2.7. Quantitative Real-Time Reverse-Transcription (Q-PCR)

Q-PCR was performed according to previously described methods [25]. Appendix A presents the sequences of the primers used for the real-time PCR experiments.

### 2.8. Western Blot Assay 

Protein extraction and Western blot analysis were performed as previously described [25]. The primary antibodies are listed in Appendix A.

### 2.9. Immunohistochemistry (IHC) Staining

GBM specimens from tissues derived from patients or xenografts from mice were fixed onto slides for IHC staining. Sample sections were blocked and incubated with the primary antibody (Appendix A), followed by staining with 200× diluted biotin-labeled secondary antibody. The percentage of positive tumor cells (0% to 100%) was determined to evaluate the predictive significance of staining markers such as STAT3 and Slug [26] by a semiquantitative scoring method.

### 2.10. Gene Expression Microarray 

For RNA preparation, total RNA was extracted from cells by using TRIzol (TRI) Reagent (Sigma, St. Louis, MO, USA) and the Qiagen RNAeasy (Qiagen, Valencia, CA, USA) column. Labeled probes were applied to a cDNA microarray containing 10,000 gene clone original cDNA fragments as previously described [22].

### 2.11. Cell Viability Assay

GBM cells were cultured on 24-well plates at a density of 1 × 10^4^ cells/well, and methyl thiazolyl tetrazolium (MTT; Sigma-Aldrich) was added to evaluate cell survival. The amount of MTT formazan, which is an indicator of cell viability, was measured in a microplate reader at a wavelength of 570 nm (SpectraMax 250; Molecular Devices, Sunnyvale, CA, USA).

### 2.12. Transwell Invasion Assay 

For the Transwell migration and invasion assay, 2 mM hydroxyurea was added to both chambers to prevent cell proliferation, and the assay was performed as previously described [22].

### 2.13. Luciferase Reporter Assay

At 36 h after transfection, the transfected cells were washed 3 times with ice-cold phosphate-buffered saline (PBS), and the luciferase reporter assay was performed as previously described [22].

### 2.14. Tumorsphere Formation Assay

GBM cells were dissociated as spheres in 24-well plates (Falcon; BD), and the tumorsphere formation assay was performed as previously described [22].

### 2.15. Immunofluorescence Staining 

Cells were cultured on chamber slides for 24 h, and immunofluorescence staining was performed as previously described [22].

### 2.16. Annexin V Apoptosis Analysis

Apoptotic cells were measured by flow cytometry using the Fluorescein isothiocyanate (FITC) annexin V and propidium iodine (PI) apoptosis detection kit I from BD Pharmingen (San Jose, CA, USA). The annexin V apoptosis procedure was performed according to previous methods [22].

### 2.17. Irradiation and Clonogenic Assay

Briefly, cells in the control group and post IR group were administered different IR doses (0, 2, 4, 6, 8, and 10 Gray (Gy)). The clonogenic assay was performed according to previously described methods [22].

### 2.18. In Vivo Red fluorescent protein (RFP) Imaging of Tumor Growth 

All animal procedures were in accordance with the institutional animal welfare guidelines and approved animal protocol of the Tri-Service General Hospital. Seven- to eight-week-old NOD-SCID female mice were acquired from the National Laboratory Animal Center (Taipei, Taiwan). The mice were intracranially (IC) injected with GBM cells in 20 µL of PBS. The skulls of the mice were immobilized in a stereotaxic apparatus for IC injection, and a ~1.5-mm hole was created in the cranium by rotating fine handheld tweezers in a circular motion. For tracing the GBM cells in the brain, we used RFP-expressing lentivirus to infect the cells before transplantation. The RFP-expressing lentivirus carries a puromycin resistant gene, which allows us to get the cells with consistent transduction efficiency. GBM cells were resuspended in 10 µl of PBS in aliquots of 1 × 10^5^ cells. The procedure was performed according to previously described methods [22].

### 2.19. Constructions of Expression Vectors

Site-directed mutagenesis of the region 1.3 kb upstream of the Slug promoter was performed using PCR-specific primers. Promoter fragments were inserted into the pGL3 reporter. 6× STAT3 RE and 6× mutSTAT3 RE were implanted into the upstream region of the minimal β-actin promoter-controlled firefly luciferase by inserting annealed 5′-phosphorylated 6× RE or 6× mutSTAT3 RE primer pairs. Detailed primer sequences are listed in Appendix A.

### 2.20. Xenograft Models and Treatments

A total of 1 × 10^5^ GBM-R2I2 cells and GBM-Par cells subjected to different therapeutic plans were transplanted into the strata of SCID mice. RFP and 3T-MR imaging Biospect system (Bruker, Ettlingen, Germany) analyses were performed according to previously described methods [22].

### 2.21. Statistical Analyses

In vitro experiments were repeated at least twice, and those experimental results were analyzed by unpaired two-sided Student’s *t*-test, Wilcoxon *t*-test, log-rank test, one-way analysis of variance (ANOVA), or chi-squared tests. A *p*-value < 0.05 was considered significant for both clinical and bench studies.

## 3. Results

### 3.1. Enhanced Tumor Motilities and Upregulated STAT3 in Radioresistant GBM Cells 

For identifying radiation effect, we established two patient-derived cell lines of (Pt1 and Pt3). The GBM-Par cells were treated by IR; then, the surviving cells were observed by invasion assay. After one treatment with IR and one invasion assay, transwell residual cells were considered the irradiated/invasive cells (R1I1). The corresponding passage-matched irradiated/invasive cell lines (R2I2-R4I4) were established (Appendix A). The Pt1-R2I2 and Pt3-R2I2 cells displayed higher radioresistance than the Pt1-R1I1 and Pt3-R1I1 cells but had similar survival fractions as Pt1-R3I3-Pt1-R4I4, and Pt3-R3I3-Pt3-R4I4 after one additional round of IR (Appendix A). Therefore, the following experiments used R2I2 cells to represent other irradiated/invasive samples. R2I2 cell lines invariably displayed higher survival fractions and invasiveness than the GBM-Par cell lines (Appendix A) in the clonogenic and invasion assays.

To search for potential downstream signals of IR-induced cell invasion, cell motility-associated gene expression levels were compared between the GBM-R2I2 and GBM-Par cells using RT^2^ Profiler PCR arrays. Cell motility-related genes were identified, with several showing increased expression (> 2.0-fold) in GBM-R2I2 relative to GBM-Par cells. The arrays showed that *STAT3*, *Rho*, and *Rac1* differed by more than two-fold in GBM-R2I2 cells compared to GBM-Par cells, and other invasive molecules such as Vimentin, N- or E-cadherin, MMP2, and MMP9 were present (Figure 1A). To confirm whether JAK/STAT3 is activated in the radio-resistant GBM cells, we used Western blotting analysis to examine the level of STAT3/JAK downstream protein, including IRF, p-STAT1, STAT1, p-JAK, and JAK (Figure 1B). In addition to STAT3/JAK signaling, we also found the EMT associated factor Slug was also increased in the radio-resistant GBM cells (Figure 1B). Taken together, we found that the radio-resistant GBM cells have activated STAT3/JAK signaling and high level of Slug expression. Stronger STAT3 activation was observed in GBM-R2I2 cells compared with GBM-Par and GBM-R1I1 cells by immunofluorescent staining (Figure 1C). Xenograft tumors from the dissected brains of injected SCID mice presented invasive characteristics including small islands and single-cell invasion of GBM-R2I2 based on histology. In contrast, GBM-Par tumors had fewer invasive phenotypes with large tumor islands, clear tumor boundaries, and a stellate appearance (Figure 1D). Importantly, GBM-R2I2 xenograft tumor specimens had higher STAT3 expression than GBM-Par specimens based on Q-PCR analysis (Figure 1E).

Collectively, these findings indicate that irradiated/invasive cells activate STAT3 expression more effectively than GBM primary cells, which are endowed with less radioresistant features, reduced invasion abilities, and lower cell motility after IR.

### 3.2. STAT3 Promotes Tumor Motility and Invasiveness through Slug

Because of the correlation between GBM invasiveness and STAT3 expression, the potential modulation of the invasive characteristics of GBM-R2I2 cells by STAT3 was investigated. Knockdown of STAT3 expression by short hairpin RNA (shRNA) constructs in the two GBM-R2I2 cell lines was achieved (Figure 2A). STAT3-knockdown in GBM-R2I2 (R2I2/sh-STAT3) cells resulted in significantly reduced invasiveness and cell motility (Figure 2B) according to the Transwell invasion assay. The correlation in GBM cells among STAT3 expression levels, EMT-like phenotypes, and cell motility suggested that STAT3 might positively regulate GBM invasion via an EMT pathway. We examined the expression levels of EMT-associated genes in three pairs of cells, i.e., GBM-Par vs. GBM-R2I2, Par/Ctrl vs. Par/STAT3, and R2I2/sh-Scr vs. R2I2/sh-STAT3, through RT^2^ Profiler PCR arrays to identify potential downstream molecular factors of the STAT3-mediated invasive phenotype. Based on the selection of EMT-related genes, expression levels (> 2.0-fold) in GBM-R2I2, Par/STAT3, and R2I2/sh-Scr cells were stronger than those in their counterparts (Figure 2C), and the array results were further corroborated by Q-PCR of GBM-Par and GBM-R2I2 cells (Appendix A). Among our screened EMT-related genes including *Slug, N-Caderin, Snail, Zeb1, Vimentin*, and others (Appendix A), the expression of transcription factor Slug in GBM-R2I2 cells were significantly higher than those in GBM-Par cells. Slug showed the most significant difference between GBM-Par and GBM-R2I2 cells (Appendix A). Taken together, the role of STAT3 in enhancing tumor motility and increasing the EMT-like characteristics in irradiated/invasive GBM cells correlated with Slug expression.

### 3.3. The STAT3/Slug Axis Modulates EMT-Like Phenotypes in Irradiated/Invasive GBM Cells

To investigate whether Slug is a regulator of STAT3-mediated EMT-like properties, we tested the causal link between STAT3 and Slug. The reduction of endogenous STAT3 decreased Slug expression, but the opposite case did not occur (Figure 2D). Reduction in STAT3 or Slug downregulated cell motility and invasive ability (Appendix A). Consistently, ectopic STAT3 overexpression increased Slug protein levels, while the ectopic overexpression of Slug had no effect on STAT3 expression or phosphorylation (Figure 2E). Moreover, cell motility and invasive abilities increased when overexpressing STAT3 or Slug compared with controls (Appendix A). In all, STAT3 upregulates Slug to activate cancer motility and invasion in GBM cells.

### 3.4. STAT3 Directly Promotes Slug Transcription

To observe the mechanism behind STAT3-mediated Slug expression, we next investigated whether STAT3 increases Slug expression via transcriptional regulation, as STAT3 is a well-known transcription factor. We searched in a 1.3-Kb sequence upstream from the Slug transcriptional start site for potential STAT3 binding sites. Three potential binding regions (RE) were identified including RE1: 5′-TTTTAGCAAAA-3′ (-1195 to -1185); RE2: 5′-TTTTTCAAAA-3′ (-472 to -463); and RE3: 5′-TTTCTTGCAAAA-3′ (-38 to -27). To determine whether the STAT3 enhancement of Slug expression is promoter region-dependent, we next generated a series of Slug promoter-driven luciferase reporter plasmids including a full-length sequence, different promoter sequence deletion lengths (D1-3), or a Slug promoter mutated at the candidate binding sites (Mut; Figure 3A, left). Following this, we cotransfected a STAT3 expression vector with the luciferase constructs in GBM cells as indicated. The luciferase reporter assays revealed that STAT3-mediated promoter activity was regulated through RE2, indicating that RE2 is the STAT3-binding site (Figure 3A, right). In further efforts to characterize the Slug promoter region dependencies of STAT3, the ChIP results for the complementary assessment of promoter activity were consistent with ChIP analysis and our reporter assays (Figure 3B). The results indicated that STAT3 enhances the activity of the endogenous Slug promoter via the same targeting regions as identified in the previous exogenous cotransfection (input, 2% of total lysate) experimental results. Empty vectors or an ectopic STAT3 expression vector were cotransfected along with firefly luciferase reporter vectors of various Slug promoter regions in GBM-Par cells. Detectable STAT3-binding signals were observed in the full-length and D1 Slug promoter vectors but not the D2, D3 or mutated Slug promoter, illustrating that STAT3 binding to the Slug promoter was dependent on the RE2 sequence within the Slug promoter sequence (Figure 3C). To further verify whether STAT3 binding positively affects transcription, we overexpressed STAT3 in the GBM-Par and GBM–R2I2 cells that was transfected with 6× STAT3 RE luciferase plasmid or mutant 6× STAT3 RE luciferase plasmid. Six repeats of the putative binding region 5′-TTTTTCAAAA-3′ identified by previous studies or six repeats of a mutated binding sequence 5′-TTTTTCAAGG-3′, was followed by a β-actin minimal promoter. The luciferase assay demonstrated that STAT3 overexpression led to a significant increase (7.3-fold) in luciferase activity in GBM-Par cells that transfected with 6× STAT3 RE plasmid (Figure 3D). Whereas this increase was not observed in the GBM-Par transfected with 6× mutSTAT3 RE plasmid. In addition, we also observed a strong increase (15.2-fold) in 6× STAT3 RE luciferase activity in GBM-R2I2 cells without STAT3 overexpression, and observed a similar increase (13.4-fold) in the GBM-R2I2 cells with STAT3 overexpression, implying that GBM-R2I2 cells have sufficient STAT3 to stimulate transcription. Consistently, we observed no significant increase in GBM-Par or GBM-R2I2 cells transfected with 6× mutSTAT3 RE plasmid. In the absence of STAT3 overexpression, only GBM-R2I2 cells can drive luciferase in the presence of 6× STAT3 RE plasmid, implying that STAT3 may involve in the transcriptional activation in GBM-R2I2 cells. Altogether, STAT3 directly promotes Slug transcription exogenously and endogenously via a specific binding sequence, 5′-TTTTTCAAAA-3′, in the RE2 region of the Slug promoter. The STAT3/Slug transcriptional regulation is a pivotal signal underlying the malignant phenotypes of GBM cells.

### 3.5. The STAT3/Slug Signal Regulates Tumor-Initiating Ability and Cancer Stem-like Properties

Previous research showed that radioresistance is often increased in CSCs [20,27]. Comparative flow cytometry analysis between GBM-Par and GBM-R2I2 cells revealed that CD133 surface levels were dramatically enhanced in GBM-R2I2 cells (Appendix A). Moreover, GBM-R2I2 cells showed higher expression levels of stemness markers, such as Bmi-1, Nanog, Sox2, Nestin, and Oct-4 (Appendix A) and showed increased sphere- formation capability (Appendix A). The increased radioresistance in our established GBM-R2I2 cells increased the potential of these cells to be CSCs.

The hypothesis that the STAT3/Slug signal enhances cancer stem-like properties and tumor-initiating ability in GBM cells was tested in the following experiments. First, the stemness transcriptome profile was examined via gene expression microarray (Figure 4A). Embryonic stem cells (ESCs) have the biologic phenotypes of stem cells, with cell specification and self-renewal [28]. GBM-R2I2 cells, Par/STAT3 cells, and Slug overexpression in R2I2/sh-STAT3 cells have increased levels and distributions of stemness factors, similar to ESCs. Multidimensional scaling (MDS) analysis and principal component analysis (PCA) showed that suppression of STAT3 in GBM-R2I2 cells diverted them away from ESC-like phenotypes; however, co-overexpression of Slug induced cells to acquire ESC-like phenotypes (Figure 4B,C).

We further explored whether the STAT3/Slug transcriptional axis mediated the tumor-initiating capability and CSC properties of GBM-R2I2 cells. STAT3 overexpression in GBM-R2I2 cells increased spheroid numbers even through several passages of the sphere-formation assay, while the simultaneous reduction of Slug decreased the STAT3-increased self-renewal (Figure 4D,E). Moreover, Slug co-overexpression rescued cells from the self-renewal inhibition caused by STAT3 knockdown (Figure 4D,E). Furthermore, R2I2/sh-STAT3 cells exhibited a significant reduction in the expression of stemness markers, while the levels of stemness markers increased in GBM-Par cells overexpressing STAT3 (Figure 4F). Orthotopically transplanted grafts of GBM clones grown in three mice showed that STAT3 overexpression enhanced tumor-initiating properties, whereas STAT3 knockdown decreased tumor initiation (Appendix A). We conclude that STAT3 is essential for forming glioblastoma and plays a substantial role in the cancer stem-like properties of radioresistant GBM.

### 3.6. Blocking the STAT3/Slug Axis Improves Radioresistance In Vitro

Given that STAT3 in irradiated/invasive GBM cells was initially identified by its upregulation in these cells, we aimed to overcome the invasiveness and radioresistance of GBM cells. Patient-derived Par, Par/STAT3, Par/Ctrl, R2I2, R2I2/sh-STAT3, and R2I2/sh-Scr cells were evaluated for colony-formation ability and cell viability 24 h after IR ranging from 0 to 10 Gy. STAT3 overexpression in Par cells increased cell survival (Figure 5A). The increase in cell viability and colony formation in R2I2/sh-Scr cells was more significant than that in R2I2/sh- STAT3 cells (Figure 5B). From the radiobiological clonogenic assays, the reduction of colonies in GBM-R2I2/sh-STAT3 cells was significantly different when treated with IR (5 Gy) compared with nontreated GBM-R2I2/sh-STAT3 cells. Co-overexpression of Slug rescued this reduction (Figure 5C, left). In contrast, the colony-formation ability of STAT3 overexpression in GBM-Par cells could not be suppressed by IR, and cotransfection of sh-Slug in GBM-Par/STAT3 cells partially overcame the radioresistance (Figure 5C, right).

STAT3 knockdown induced significantly more apoptotic GBM-R2I2 cells after 5 Gy IR, whereas Slug co-overexpression reduced the percentage of Annexin V positive cells (Figure 5D,E left). In contrast, STAT3 overexpression caused GBM-Par cells to escape from apoptosis after IR, while the co-knockdown of Slug dramatically increased the number of apoptotic cells (Figure 5D,E right). Therefore, inhibition of the STAT3/Slug axis overcame radioresistance through avoiding apoptosis.

### 3.7. Blocking the STAT3/Slug Axis Synergistically Enhances the Efficacy of Radiosensitivity and Improves Survival in a GBM-R2I2 Xenograft Model

We evaluated the effects of STAT3/Slug signaling on radiotherapy in vivo. Histology analysis of xenograft brain tumors revealed that the pathologies of GBM-R2I2/sh-STAT3 tumors after 5 Gy RT exhibited dramatically reduced tumor size in 3T MRI compared with those of GBM-R2I2/sh-Scr tumors post-RT (5 Gy) (Figure 6A). RFP imaging indicated that the tumor volumes in xenotransplanted GBM-R2I2/sh-STAT3 mice were significantly shrunken compared with those of GBM-R2I2/sh-Scr-injected mice after 5 Gy RT (Figure 6B). The combination of RT and STAT3 knockdown had a synergistic effect, significantly reducing tumor volumes (Figure 6C). 

In contrast, GBM-Par/Ctrl-transplanted mice showed slowly growing tumors post-RT, and ectopic STAT3 increased the growth rate of GBM-Par/Ctrl cells (Appendix A). Q-PCR analysis demonstrated that Sox2 and Oct4 expression levels were significantly reduced in xenograft sections from GBM-R2I2/sh-STAT3-transplanted mice compared with those from mice subjected to other treatments (Appendix A). Sh-STAT3 in combination with IR (5 Gy) prolonged the survival of GBM-R2I2 intracranial tumor-bearing mice (Appendix A). Moreover, ectopic STAT3 expression resulted in reduced survival rates in GBM-Par-transplanted mice compared with those of untreated GBM-Par-bearing mice (Appendix A). The STAT3/Slug axis modulates the resistance to radiation and CSC properties in vivo.

### 3.8. STAT3/Slug-Expressing Cells Are Higher in Recurrent Human Samples of GBM 

Following the results of the in vivo and in vitro experiments, we analyzed STAT3 expression in nine GBM patient samples (Appendix A) using IHC staining. The grading of Slug closely correlated with STAT3 expression in patient samples in IHC staining (Figure 6D). All patients received the standard therapy of concurrent chemoradiotherapy following their first operation, but five patients (Pt 1, 3, 5, 7 and 8) had cancer relapse and underwent a second surgery. Compared with the first surgical samples, the percentages of STAT3-positive cells were significantly increased in patients 1, 3, 5 and 8 (Appendix A). Therefore, STAT3/Slug staining may associate GBM recurrence. Based on the closely associated relationship between these two molecules in human samples, we found colocalization between STAT3 and Slug in the same foci of GBM tissue from Pt1, which was STAT3^hi^ and Slug^hi^ (Figure 6E). In conclusion, STAT3/Slug regulates radioresistance and induces tumorigenesis, tumor invasion, and CSC properties (Figure 6F).

## 4. Discussion

The upregulation of GBM radioresistance by STAT3 has recently been described [18]. To the best of our knowledge, this is the first study to reveal that STAT3 directly regulated Slug transcription and induced radioresistance in GBM. Slug is an important factor in STAT3-dependent enhancement of radioresistance, motility, and CSC properties based on the following findings: (i) STAT3 directly bound to the Slug promoter; (ii) STAT3 upregulated Slug and contributed to tumor invasion, radioresistance, and CSC properties; (iii) the coexpression of STAT3 and Slug in GBM enhanced radioresistance and correlated with poor prognosis.

STAT3 has biological consequences in some malignancies and plays a critical role in activating tumor invasiveness outside the CNS [29]. Activation of JAK2/STAT3 signaling is associated with enhanced radioresistance of hepatocellular carcinoma cells [30]. STAT3 phosphorylation can suppress radioresistance and improve outcome in glioblastoma stem cells [31]. Consistent with these reports, our present findings indicate that STAT3 expression is a crucial molecular factor linking radioresistance, tumor motility, and cancer stem-like properties in GBM cell lines.

Matrix metalloproteinases (MMPs) have been implicated as playing an important role in mediating GBM cell invasion [32]. MMPs are also one of the STAT3 downstream signals regulating tumor migration and invasion [33]. Previous research has shown that matrix metalloprotease-2 (MMP-2) and MMP-9 expression are significantly elevated in high-grade gliomas [34]. MMP-2 and MMP-9 mediate GBM cell motility to transfer latent transforming growth factor β (TGF-β) into active form, which in turn induces MMP-2 in a feedback loop [35]. The activity of MMP-9 expression could be regulated by the activation of STAT3 and TGF-β [36]. Tumor invasion begins with cell migration, a process that is driven by focal adhesion assembly and cytoskeleton rearrangement [37,38]. Several studies have shown that Slug plays an important role in the EMT phenotype of multiple malignant cancer types outside the CNS [39,40]. MMP-2 and MMP-9 are also regulated by Slug. Our results indicated higher expression of Slug in GBM-R2I2 cells than in GBM-Par cells. Moreover, the reduction of Slug suppressed the invasiveness of GBM-R2I2 cells.

Mesenchymal-epithelial transition (MET) and EMT have been closely linked to “stemness” in tumorigenesis. MMPs, epithelial cadherin (E-cadherin), neural cadherin (N-cadherin), and vimentin are the main proteins involved in the transformation between EMT and MET [41]. The radioresistance of GBM with CSC characteristics is particularly associated with their intrinsic ability to efficiently activate the DNA damage response [42,43]. A previous study showed that Slug directly induces Sox2 activation and increase tumor-initiating cell properties [44]. In our study, ectopic expression of Slug rescued the effect of shSTAT3 on repressed stem-like properties, including tumor-initiating capabilities, in GBM-R2I2/sh-STAT3 cells.

## 5. Conclusions

In conclusion, our study showed that activating STAT3/Slug signaling induced radioresistance, increased EMT-like phenotypes, such as migration and invasion abilities, and enhanced the acquisition of stem-like cell characteristics of GBM cells. Our findings imply that STAT3/Slug signaling has a major role in GBM invasion, radioresistance, and recurrence, suggesting that this pathway may be a crucial target for GBM therapy. We believe that the STAT3/Slug axis could be a latent therapeutic target to suppress GBM invasion and CSCs as well as to enhance the synergistic effects of IR therapy. This study provides insights into the development of potential treatments that may be able to overcome radioresistance.

## Figures and Tables

**Figure 1 cancers-10-00512-f001:**
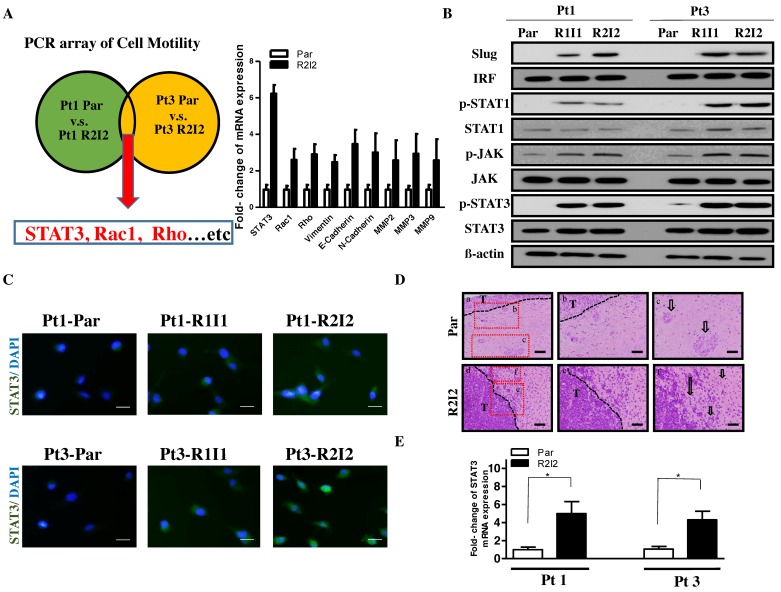
Radioresistant Glioblastoma multiforme (GBM) cells display higher Signal transducer and activator of transcription 3 (STAT3) expression. (**A**) Left panel: Schema for identifying motility-related genes using the Human Cell Motility RT^2^ Profiler PCR Array. Right panel: Quantitative Real-Time Reverse Transcription (Q-PCR) analysis of invasion-related genes. (**B**) Western blot analysis showing that STAT3/JAK signaling protein and EMT-associated protein Slug in GBM-Par and GBM-R2I2 cells from two individual patients. (**C**) Immunofluorescence of STAT3 expression in GBM–R2I2 and Par cells. DAPI (4′,6-diamidino-2-phenylindole) staining identified the nuclei. Scale bars: 20 μm. Blue: DAPI; Green: STAT3. (**D**) Upper panel: GBM-Par histology showing reduced cell invasive characteristics such as a clear tumor border (**b**) and large mass islands (**c**; arrow) with a stellate appearance. Lower panel: GBM-R2I2 cells have invasive properties such as an indistinct tumor borderline (**e**) and small tumor islands (**f**; arrow), along the white matter tracts with single-cell invasion and invasion as cell clusters. Scale bars: 200 μm (**a** and **d**), and 100 μm (**b**, **c**, **e** and **f**). T: Main tumor mass. (**E**) GBM-R2I2 cells show higher STAT3 expression levels than those isolated from GBM-Par cells of xenograft tumors as determined by Q-PCR analysis. * *p* < 0.01 by Student’s *t* -test. The data shown are the mean ± SD of three independent experiments.

**Figure 2 cancers-10-00512-f002:**
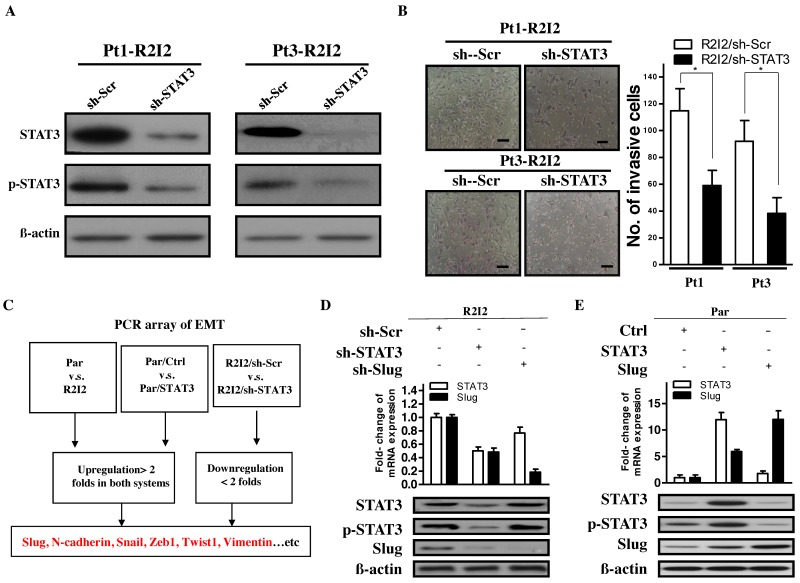
Slug is the downstream target of STAT3-modulated cell invasion. (**A**) STAT3-knockdown cell lines were generated in previously isolated Pt1-R2I2 and Pt3-R2I2 cells. Western blotting was used to analyze the expression or knockdown efficiency of STAT3. (**B**) Pt1-R2I2 cells subjected to transfection with the scrambled shRNA control vector (Pt1-R2I2/sh-Scr) or the sh-STAT3 vector (Pt1-R2I2/sh-STAT3) and Pt3-R2I2 cells transfected with the scrambled shRNA control vector (Pt3-R2I2/sh-Scr) or the sh-STAT3 vector (Pt3-R2I2/sh-STAT3) were subjected to a wound-healing assay. Scale bars: 50 μm. * *p* < 0.05 (**C**) Schematic diagram for analyzing epithelial-mesenchymal transition (EMT)-related factors using the Human Epithelial to Mesenchymal Transition RT^2^ Profiler PCR Array. (**D**) Expression of STAT3, p-STAT3, and Slug in GBM-R2I2 cells. β-actin was used as a loading control. (**E**) STAT3, p-STAT3, and Slug expression in GBM-Par cells. β-actin was used as a loading control.

**Figure 3 cancers-10-00512-f003:**
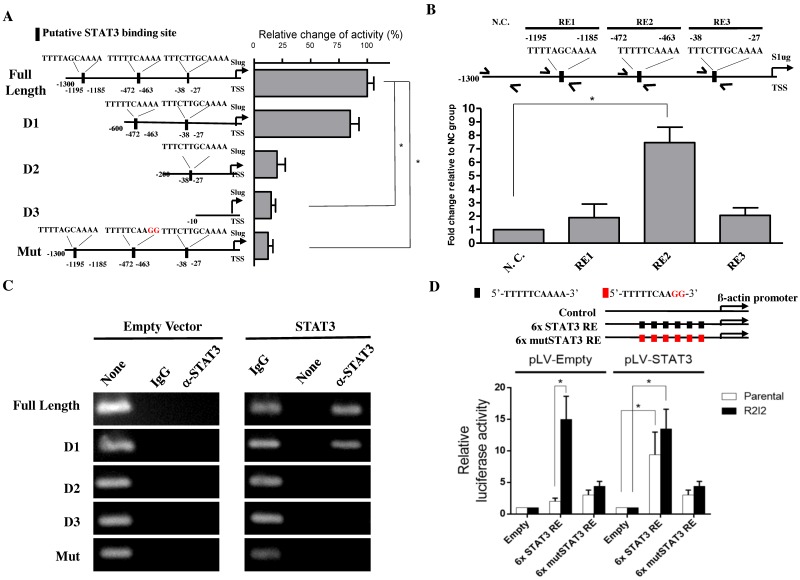
STAT3 directly regulated Slug transcription via binding to the STAT3 responsive sequence in the Slug promoter. (**A**) Left side: A graphical illustration of the full-length, deleted, and mutated Slug promoter-driven reporter plasmids. Right side: The quantification of luciferase activity driven by STAT3 from the different promoters in GBM-Par cells. (**B**) STAT3 at the indicated loci in the Slug promoter, including the nonspecific control (N.C.), RE1, RE2, and RE3 analyzed by Quantitative Chromatin Immunoprecipitation (Q-ChIP) assay in GBM-R2I2 cells. The amplicon N.C. was selected according to STAT3-upstream which has no STAT3 binding sites. (**C**) ChIP assay of STAT3 at the full-length or mutated Slug promoter regions. Specific STAT3-binding signal after ectopic STAT3 expression was compared with empty vector in GBM-Par cells. IgG antibody was used as a negative control. (**D**) Luciferase assay of examining the transcriptional effect induced by STAT3 overexpression in the Par or R2I2 GBM cells. The luciferase plasmid carrys a β-actin promoter, which contains empty, six repeated wild type or mutated STAT3 RE on the plasmid. Values expressed as fold enrichment relative to its empty group. All data represented as mean ± SD, *n* = 3, * *p* < 0.01.

**Figure 4 cancers-10-00512-f004:**
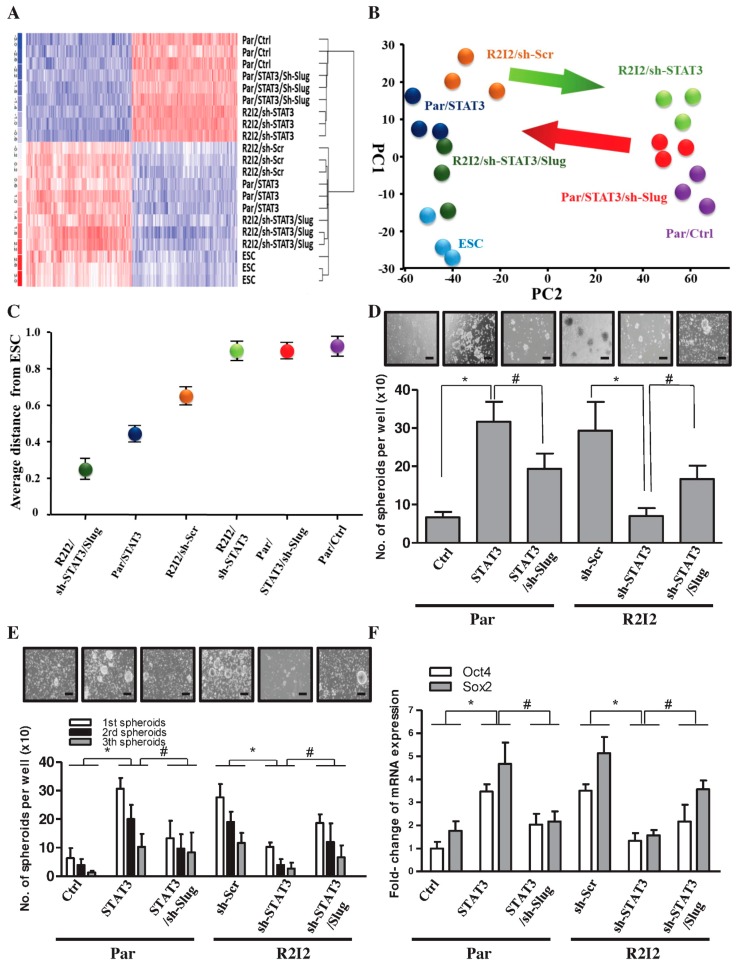
STAT3/Slug signaling mediates stemness properties in GBM-R2I2 cells. (**A**) Gene tree (genetic microarray) showing differentially expressed targets in human embryonic stem cells (ESC), GBM-Par/Ctrl cells, GBM-Par/STAT3 cells, GBM-Par/STAT3/sh-Slug cells, GBM-R2I2/sh-Scr cells, GBM- R2I2/sh-STAT3 cells, and GBM-R2I2/sh-STAT3/Slug cells as indicated by a hierarchical heat map. Blue: low gene expression, Red: high gene expression. (**B**) Principal component analysis (PCA) and (**C**) Multidimensional scaling (MDS) analysis demonstrated the average lineage transcriptome distances between the seven cell lines. (**D**) Sphere formation and (**E**) self-renewal assays were used to analyze sphere-forming rate in the above six cell lines. Scale bars: 50 μm. (**F**) Q-PCR expression of Sox2 and Oct4 in the above six cell lines. * *p* < 0.01; ^#^
*p*
*<* 0.01 by Student’s *t*-test. The data are presented as the mean ± SD of three independent experiments.

**Figure 5 cancers-10-00512-f005:**
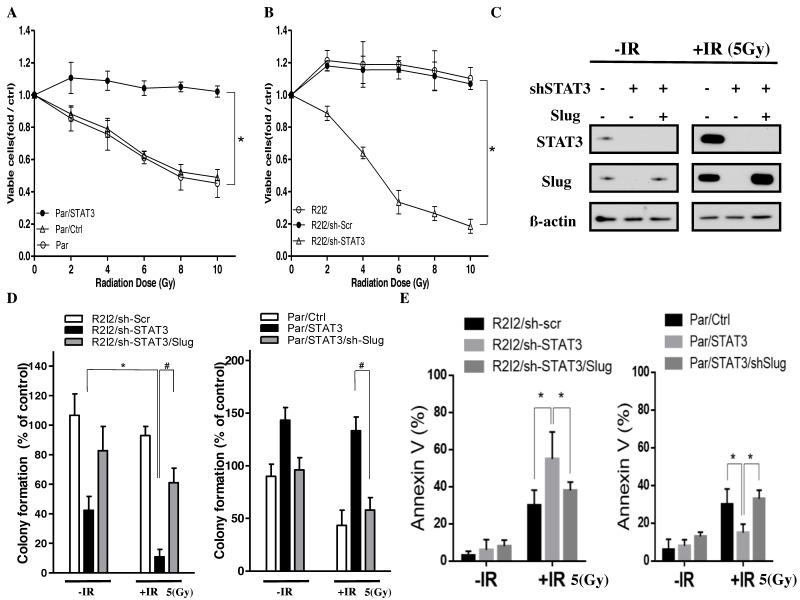
Silencing of STAT3/Slug signaling in combination with ionizing radiation (IR) suppresses cell viability. (**A**) IR doses ranging from 0 to 10 Gy were given to GBM-Par cells with sh-Scr or sh-STAT3, and (**B**) GBM-R2I2 cells with sh-Scr or sh-STAT3. (**C**) Western blotting of GBM cells that were treated with sh-Scr, sh-STAT3, or sh-STAT3/Slug with or without IR (5 Gy). β-actin was used as loading control. (**D**) Colony-forming ability and (**E**) Annexin V staining of GBM-R2I2 cells that were treated with sh-Scr, sh-STAT3, or sh-STAT3/Slug with or without IR (5 Gy); in addition, GBM-Par cells with vector control, ectopic expression of STAT3, or ectopic expression STAT3/sh-Slug were treated with or without IR. * *p* < 0.01; ^#^
*p*
*<* 0.01 by Student’s *t*-test. The data are presented as the mean ± SD of 3 independent experiments.

**Figure 6 cancers-10-00512-f006:**
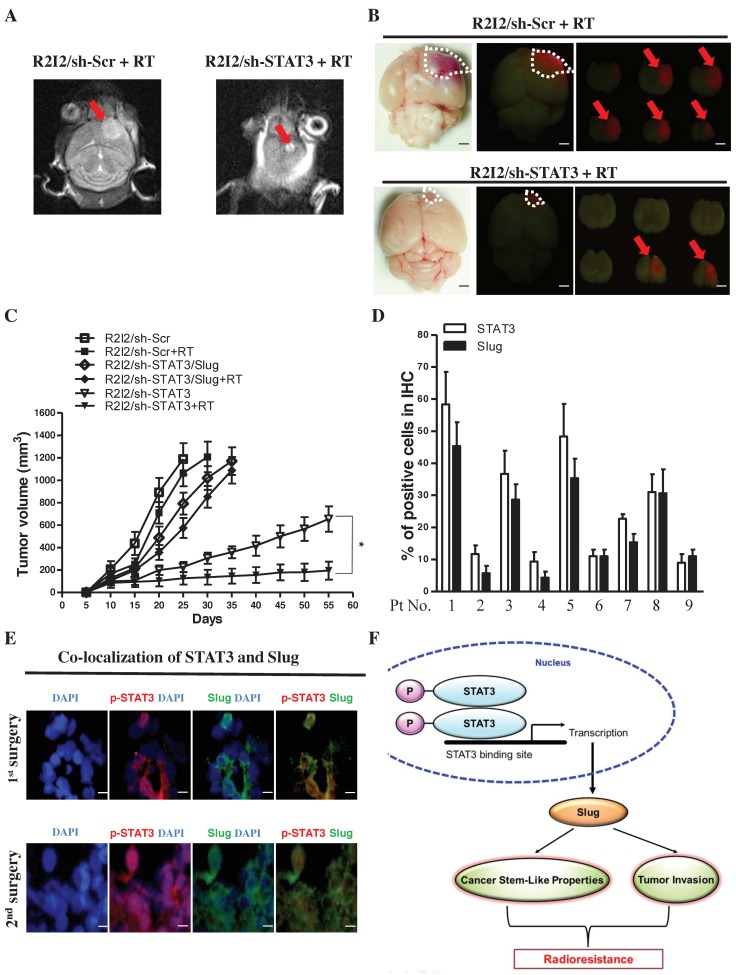
Suppressing the STAT3/Slug axis shows synergistic effects on radiosensitivity in vivo. (**A**) GBM-R2I2 cells were subinjected into NOD-SCID mice intracranially. 3T MRI and (**B**) RFP images show tumor volumes in GBM-R2I2/sh-STAT3 tumor-bearing mice compared with GBM-R2I2/sh-Scr mice after IR (5 Gy). Scale bars: 50 μm (**C**) The size of tumor volumes in GBM-R2I2 mice treated with sh-STAT3 + 5 Gy IR were significantly smaller than those receiving the other protocols. (**D**) Immunohistochemistry (IHC) staining detected STAT3 and Slug protein in nine GBM patient samples. (**E**) STAT3 and Slug protein colocalization in tissues from Pt1 by immunofluorescent staining. Scale bars: 20 μm. Blue: DAPI; Green: Slug; Red: p-STAT3. (**F**) Schematic illustration depicting the STAT3/Slug axis associations with cancer stemness and tumor invasion that induce resistance to radiation.

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
