# Peer review of "The STAT3/Slug Axis Enhances Radiation-Induced Tumor Invasion and Cancer Stem-like Properties in Radioresistant Glioblastoma"

_cancers, 2018, doi:10.3390/cancers10120512_

Round 1

Reviewer 1 Report

In the supplementary table1, please check the legends. And if Pt1, 2, 3 means the origin of tumor samples, the cell line is not mentioned anywhere in the manuscript. If then, it is better to delete the data from the Pt-2 derived tumors.  And "Mice" is not appropriate description.

What is XXXX in the flollowing description?

In vivo RFP imaging of tumor growth
All procedures involving animals were in accordance with the institutional animal welfare
guidelines and approved animal protocol of the XXXX.  --

Author Response

Response to Reviewer 1 Comments

Point 1: In the supplementary table1, please check the legends. And if Pt1, 2, 3 means the origin of tumor samples, the cell line is not mentioned anywhere in the manuscript. If then, it is better to delete the data from the Pt-2 derived tumors.  And "Mice" is not appropriate description.

Authors’ Response: We thank you for your comments and questions. We have deleted the data from the Pt-2 derived tumors. We also revise "Mice" to "Pt No.".

Point 2: What is XXXX in the flollowing description?

In vivo RFP imaging of tumor growth

All procedures involving animals were in accordance with the institutional animal welfare guidelines and approved animal protocol of the XXXX.

Authors’ Response: We thank you for your comments and questions. We have revised XXXX by Tri-Service General Hospital.

Reviewer 2 Report

This revised version is acceptable for the publication of 'Cancers'.

Author Response

Response to Reviewer 2 Comments

Point 1: This revised version is acceptable for the publication of 'Cancers'.

Authors’ Response: We thank you for your comments and questions.

Reviewer 3 Report

Most of my comments have been adequately addressed. Regarding my first suggestion (Figure 3 / use IR to increase STAT3 activity rather than overexpression), the experiment the authors added is not the one I suggested, but it is satisfactory and improves the figure. Two minor issues remain uncorrected. 1. For the Kaplan-Meier curves in supplemental figure 4, the authors were asked to use the log rank test instead of a t-test to compare curves. They stated in the reply that they did this, but the legend to the figure still says a t-test was performed. This must be corrected. 2. Regarding the question about whether the RFP expression was the same for all cell lines they compared, the authors replied that RFP+ cells were selected by puromycin. This should be included in the methods.

Author Response

Response to Reviewer 3 Comments

Comments and Suggestions for Authors

Most of my comments have been adequately addressed. Regarding my first suggestion (Figure 3 / use IR to increase STAT3 activity rather than overexpression), the experiment the authors added is not the one I suggested, but it is satisfactory and improves the figure. Two minor issues remain uncorrected.

Point 1: For the Kaplan-Meier curves in supplemental figure 4, the authors were asked to use the log rank test instead of a t-test to compare curves. They stated in the reply that they did this, but the legend to the figure still says a t-test was performed. This must be corrected.

Authors’ Response: We thank you for your comments and questions. We have corrected the log rank test instead of a t-test in supplemental figure 4 legend.

Point 2: Regarding the question about whether the RFP expression was the same for all cell lines they compared, the authors replied that RFP+ cells were selected by puromycin. This should be included in the methods.

Authors’ Response: We thank you for your comments and questions. We have added < For tracing the GBM cells in the brain, we used RFP-expressing lentivirus to infect the cells before transplantation. The RFP-expressing lentivirus carries a puromycin resistant gene, which allows us to get the cells with consistent transduction efficiency. > in the methods.
